# Retroperitoneal Sarcoma Care in 2021

**DOI:** 10.3390/cancers14051293

**Published:** 2022-03-02

**Authors:** Erika Schmitz, Carolyn Nessim

**Affiliations:** 1Faculty of Medicine, University of Ottawa, Ottawa, ON K1H 8M5, Canada; eschmitz@toh.ca; 2Department of General Surgery, Division of Surgical Oncology, The Ottawa Hospital, Ottawa, ON K1H 8L6, Canada; 3The Ottawa Hospital Research Institute (OHRI), Cancer Therapeutics Program, Ottawa, ON K1H 8L6, Canada

**Keywords:** retroperitoneal sarcoma, sarcoma, soft-tissue tumor, radiation therapy, systemic therapy

## Abstract

**Simple Summary:**

Soft-tissue sarcomas are exceedingly rare, accounting for <1% of all adult malignancies. Sarcomatous tumors are located within the retroperitoneum in <10% of cases. International collaborations have been instrumental in advancing our understanding of these rare tumors in the last decade. Notably, standard treatment has become upfront primary surgical resection for most subtypes. Radiation and systematic therapy have a limited role. In this 2021 review, we detail the anatomical boundaries of the retroperitoneum, risk factors including genetic predispositions, clinical characteristics, histologic-specific management, contemporary standard of care, recent advancements and limitations of knowledge in the treatment of retroperitoneal sarcoma care.

**Abstract:**

Soft-tissue sarcomas are biologically heterogenous tumors arising from connective tissues with over 100 subtypes. Although sarcomas account for <1% of all adult malignancies, retroperitoneal sarcomas are a distinct subgroup accounting for <10% of all sarcomatous tumors. There have been considerable advancements in the understanding and treatment of retroperitoneal sarcoma in the last decade, with standard treatment consisting of upfront primary surgical resection. The evidence surrounding the addition of radiation therapy remains controversial. There remains no standard with regards to systemic therapy, including immunotherapy. Adjunctive therapy remains largely dictated by expert consensus and preferences at individual centers or participation in clinical trials. In this 2021 review, we detail the anatomical boundaries of the retroperitoneum, clinical characteristics, contemporary standard of care and well as recent advancements in retroperitoneal sarcoma care. Ongoing international collaborations are encouraged to advance our understanding of this complex disease.

## 1. Introduction

Soft-tissue sarcomas are biologically heterogenous tumors arising from mesenchymal cells, notably fat, nerves, blood vessels and connective tissues with over 100 subtypes [1]. Although sarcomas account for <1% of all adult malignancies, retroperitoneal sarcomas are a distinct subgroup accounting for <10% of all sarcomatous tumors. There is no sex preponderance, and peak incidence is often the fifth decade of life. There have been considerable advancements in the understanding and treatment of retroperitoneal sarcoma in the last decade. In this 2021 review, we detail the anatomical boundaries of the retroperitoneum and introduce the concept of compartmental resection and surgical approach, clinical characteristics of the most common soft-tissue sarcomas of the retroperitoneum, as well as contemporary standard of care and recent advancements in retroperitoneal sarcoma care. For the purposes of completeness, this review will focus solely on retroperitoneal sarcomas (RPS) excluding pediatric sarcomas, non-sarcomatous retroperitoneal tumors, desmoid tumors, gastrointestinal stromal tumors, and extremity sarcomas.

## 2. Anatomy

Approximately 15% of all sarcomas are located within the retroperitoneum [2,3]. The retroperitoneum is broadly defined as an anatomic space located posterior to the peritoneal cavity. It is broadly bordered by the diaphragm superiorly, the iliopsoas inferiorly, the paraspinous muscle medially, the transversalis fascia of the abdominal wall laterally, and the psoas, quadratus lumborum, iliacus and transverse abdominis muscle posteriorly. The retroperitoneum space is divided into 3 compartments, notably the greater vessel compartment, the posterior compartments and the lateral compartment. The latter is sub-divided into three spaces, notably the anterior (APS), posterior pararenal (PPS) spaces and the perirenal space (PS) illustrated in Figure 1. The borders, and solid organ contents of each compartment are described in Table 1. It is worth noting in addition to these solid organs, the retroperitoneal space contains a complex network of lymphatic channels, nerves, vessels, as well as connective tissue and fat.

Sarcomatous tumors that arise in the retroperitoneum often spread within the retroperitoneal planes, and may invade or abut multiple retroperitoneal structures rendering multi-visceral resections a commonality. Consequently, a thorough knowledge of the retroperitoneal compartments, association with major vessels and organs, planes of dissection and local extension, communication routes with the extra-retroperitoneal spaces, as well as their associated radiologic appearances, is relevant to the surgeon [4,5,6,7]. Details regarding the compartmental surgical management of RPS are found below.

## 3. Histologic Subtypes

Sarcomatous lesions are a broad category of tumors which encompass at least 100 distinct histologic subtypes, classified typically accordingly to their cell type of origin. With advancements in molecular analysis techniques, the WHO classification was mostly recently updated in 2020 by expert pathologists in collaboration with geneticists, radiologists, surgical oncologists, and medical oncologists [1]. The new classification scheme still classifies tumors based on cell of origin, while newly integrating advanced molecular profiling and genetic techniques [8].

Categorization of mesenchymal tumors is known as one of the most challenging fields within pathology [8]. Due to tumor rarity, tumor complexity, technological complexity, and educational expertise, there are overall high rates of diagnostic inaccuracy compared to other tumor types [8]. Consequently, it has become standardized practice to obtain second pathologic opinion in expert centers.

The most common subtypes of sarcoma within the retroperitoneum in descending order of frequency: liposarcoma, characterized as either dedifferentiated liposarcoma (DDLPS) or well-differentiated liposarcoma (WDLPS), leiomyosarcoma (LMS), undifferentiated sarcoma (US), solitary fibrous tumor (SFT) and malignant peripheral nerve sheath tumor (MPNST), and fibrosarcoma (FS) [9]. Malignant fibrous histiocytomas were historically the most common entity within the retroperitoneum; however with advances in histopathology, such tumors have been re-evaluated to represent leiomyosarcomas or dedifferentiated sarcomas [10]. The average prevalence of each subtype was estimated by Carbone et al. after reviewing 8 of the largest case series [11] and illustrated in Figure 2. To add to their complexity, each subtype displays different behavior, making these tumors exceedingly difficult to treat [6,9,12,13,14].

## 4. Genetics, and Familial Syndromes

Although most sarcomas are believed to be sporadic in nature, a variety of familial syndromes are known to be contributory to the development of sarcoma [15]. However, two major studies have questioned the sporadic nature of sarcomas in recent years. The International Sarcoma Kindred Study (ISKS) screened 1192 apparently sporadic sarcoma patients for 72 known cancer-associated germline mutation component and discovered that 55% of cases carried notable pathologic variants [16]. This finding was verified in a cohort of 66 Southern Asian patients under 50 years of age, where 13.6% of cases harbored at least one germline mutation in a panel of 52 known cancer-associated genes [17]. These studies do raise the hypothesis that a significant proportion of sarcomas actually do have an underlying hereditary component, and question of whether young adult sarcoma patients should be considered for genetic testing. Clinicians should typically obtain a thorough familial history at time of consultation for a new sarcoma, and consider hereditary predisposition when offering radiation or chemotherapy due to heightened radiation-induced and second–primary malignancies [15,18,19]. Nevertheless, most known syndromes are related to embryologic and extremity sarcomas [15]. These include but are not limited to Beckwith–Wiedermann, Bloom, Constitutional Mismatch repair syndrome, Costello, Familial pleuropulmonary blastoma, Gorlin syndrome, Noonan, Rubinstein–Taybi, Rothmud–Thrompson syndrome II, Werner, and Mosaic variegated aneuploidy [15]. Gastrointestinal tumors are related to Carney–Stratakis, Familial GIST syndrome and Neurofibromatosis 1 [15]. Desmoid tumors are related to Familial Adenomatous Polyposis Syndrome [20]. Soft-tissue sarcomas (STS)-associated syndromes are summarized in Table 2. The relatively more common syndromes of Neurofibromatosis Type 1, TP53-related cancer syndrome, and Hereditary retinoblastoma (HR), which are detailed below. Of note, no studies to date have outlined the proportion of retroperitoneal sarcomas among those with genetic syndromes. Additional common genetic syndromes with unclear association with STSs include Lynch Syndrome [21].

### 4.1. Neurofibromatosis Type I (NF1, von Recklinghausen’s Disease)

Neurofibromatosis type I arises from an autosomal dominant loss-of-function mutation of the NF1 gene for the neurofibromin tumor-suppressor protein, located on chromosome 17q11.2. The NF1 gene protein is expressed primarily in neuronal, glial and melanocytic cells with mutations accounting for the typical widespread neurofibromas, and cutaneous manifestations including skin café-au-lait lesions, axillary and groin freckling, optic gliomas, Lisch nodules, skeletal deformities and learning disabilities [24]. Associated sarcomatous malignancies include gastrointestinal stromal tumors, rhabdomyosarcoma, and 8–13% lifetime risk of development of malignant peripheral nerve sheath tumors (MPNST, [25]). Although there does not appear to be a clear genotype–phenotype correlation, 5–10% of NF1 patients have microdeletion mutations with associated overall more severe manifestations, compared to the usual NF1 patients with small truncating mutations [26]. It is estimated that NF1 patients with microdeletions have a 16–26% lifetime risk of developing MPNST, with clinical implications being a lowered threshold of investigations for nonspecific complaints [26].

### 4.2. TP53-Related Cancer Syndrome (hPT53c, Li–Fraumeni)

TP53-related cancer syndrome arises from an autosomal dominant loss-of-function mutation most associated with gene TP53 for nuclear protein transcription factor located on chromosome 17q13.1 but has also been reported as a TP53 mutation on 1q23 and a checkpoint kinase CHEK2 on 22q12., which regulates P53. The p53 protein primarily plays a role in cell cycle, DNA repair and apoptosis. Historically known as Li–Fraumeni Syndrome, and SBLA syndrome due to its association with sarcomas, breast, leukemia, adrenal gland cancers before the age of 45, the syndrome has been expanded to the TP53-related cancer syndrome (hPT53c) due to its associated with a myriad of other malignancies [19]. A key feature of the classic LFS criteria was the development of a sarcoma prior to the age of 45, with the modified Chompret criteria accounting for the additional associated malignancies [22]. Sarcomas are one of the most common cancers associated with TP53 mutations, and was reported as occurring in 27% of carriers [22]. The reported subtypes in decreasing frequency including rhabdomyosarcoma, leiomyosarcoma, liposarcoma, and fibrohistocytic tumors [22]. The clinical implications with regards to care of the sarcoma patient with a TP53 mutation are individualized risk assessment, consideration of chemotherapy and radiation-sparing approaches, reproductive counselling, and surveillance protocols [19,27].

### 4.3. Hereditary Retinoblastoma

Hereditary retinoblastoma arises from an autosomal dominant mutation in the RB1 tumor-suppressor gene for Rb cell cycle regulator. Most patients with HR develop bilateral ocular retinoblastomas by 18 months of age. Survivors often develop second–primary tumors within 10–50 years, with bone sarcoma accounting for 25–30% of tumors and STS for 12–32% [23]. The clinical importance includes screening for childhood malignancies at the time of consultation for sarcoma, and surveillance for synchronous or metachronous melanomas and epithelial tumors [28].

## 5. Additional Risk Factors

### 5.1. Radiation-Induced Sarcoma (RSTS)

Although most sarcomas do not have any identifiable risk factors, radiation exposure has been recognized since the 1920s to induce sarcomas [29]. Although the interval between exposure and soft-tissue sarcoma development has historically been reported as at least 5 years, more recent retrospective reviews have shown that the latency period can be shorter than 3 years [30]. There does tend to be a radiation dose correlation with the incidence of STS development, and STS often occur at the margins of the radiation fields [31,32]. Overall, RSTS tend to be more aggressive with higher predilection for local recurrence and metastases as compared to sporadic STS [33,34].

### 5.2. Other Risk Factors

Additional environmental toxins and host-related immunosuppressive etiologies have been outlined in a review [32]. Population studies have discovered a higher preponderance of sarcoma within African American population [35], and the Māori population of New Zealand [36]. Further studies are required in order to determine the association with determinants of health, such as socioeconomic status.

## 6. Clinical Presentation

Due to their retroperitoneal origin within large potential spaces, tumors can often reach an impressive size without causing any signs nor symptoms. The tumor eventually reaches an average size of 15 cm when symptoms develop, which are often non-specific abdominal complaints, including abdominal fullness, early satiety with associated malnutrition, and a palpable mass. Patients often attribute increased abdominal girth and tumor mass to weight gain and will attempt weight loss prior to presentation. In addition, visceral obesity is common among patients with RPS [37], and may contribute to a recognition delay [38]. Uncommonly, patients present with obstructive symptoms of the gastrointestinal or genitourinary tract, venous return, or hemorrhage. Cases have been reported of first presentations with pancreatitis [39,40], ischemic colitis [41], paraneoplastic syndromes [42], as well as incidental tumors found intra-operatively. Most often, tumors are detected incidentally on imaging for an unrelated complaint.

## 7. Investigations

Adequate imaging is paramount in determining the appropriate management strategy, notably resectability, in RPS. Contrast enhanced computed tomography (CT) is the most widely available and useful test in determining organ of origin and soft-tissue composition [43]. Typical associated findings include the displacement of retroperitoneal organs, infiltration of retroperitoneal organs, among others. It is also used for staging of the chest, abdomen, and pelvis for evidence of metastatic nodal and solid organ disease. Magnetic resonance imaging (MRI) is primarily reserved for patients with allergy to iodinated contrast agents, diagnostic uncertainty, involvement of critical structures not well visualized on CT, such as structures involving the pelvis spine or major nerves. In addition, MRI is useful for delineating the extent of peritumoral edema prior to initiation of radiation therapy [44]. Positions Emissions Tomography (PET) has no routine role in sarcoma care. However, PET has been shown to differentiate high-grade from low-grade lesions, but not low-grade from benign lesions [45,46]. PET can also rule out distant metastatic disease in high-grade tumors if there is a doubt on CT scan. A diagnosis of well-differentiated liposarcoma (WDLPS) can be accurately made based on radiologic findings alone, which likely explains why lower rates of RPB were observed in these patients in comparison to those with non-WDLPS (24).

## 8. Percutaneous Biopsy

Proof of histology with biopsy is routinely recommended in cases of retroperitoneal sarcoma to facilitate optimal perioperative management after discussion with a multidisciplinary tumor board. In addition, it is also useful to clear any doubt that the tumor may truly represent a sarcoma as opposed to a benign, metastatic or non-sarcomatous malignant etiology where management may differ.

A CT- or US-guided approach with at least 4–5 passes with a 14 G or 16 G sheathed core needle retroperitoneal biopsies (RPB) are recommended to facilitate immunohistochemistry processing [43], as opposed to fine needle aspirate, in order to yield adequate tissue sample for analysis. Imaging guidance is recommended in order to sample the most solid and dedifferentiated component, rather than areas of cysts or necrosis [43] while avoiding critical vascular structures, notably the aorta, inferior vena cava and renal hilum. A retroperitoneal approach is favored as opposed to an intraperitoneal or surgical approach, due to theoretical risk of peritoneal seeding, distortion of tissue planes precluding eventual wide local resection, subjecting the patient to operative risk, as well as additional biological effects on tumor progression [47]. If detected incidentally at the time of unrelated abdominal surgery, interval image-guided RPB would still be favored rather than intra-operative peritoneal biopsies.

Concerns regarding retroperitoneal risks particularly involve acute complications, needle track seeding (NTS) and oncologic outcome, especially given that the track is typically left in situ with standardized intraperitoneal resections. However, such risks are overall quite low, especially while using the coaxial technique. Complications of RPB are reported at 3.1%, of which the majority were intra-tumoral hemorrhage [48]. The range of NTS has been reported as being between 0.37 and 2% in retrospective reviews of major international sarcoma centers [48,49,50]. In addition, RPB overall does not influence the oncologic outcome, notably overall survival [51,52]. RPB has been indirectly associated, given the ability to determine sarcoma subtype, administration of neoadjuvant therapy, subsequent complete tumor resections and thus indirectly an overall survival advantage [52]. Consequently, biopsy is recommended in all major retroperitoneal sarcoma treatment guidelines (NCCN, TARPSWG, ESMO) [53,54,55].

## 9. Staging and Nomograms

The most widely used and recommended staging system for retroperitoneal sarcoma is the American Joint Committee on Cancer (AJCC), 8th edition (2017, [56]). The AJCC 8th edition stratifies stage according to primary tumor size, presence of regional lymph nodes, distant metastasis, and grade, the latter of which corresponds to a sum of scores related to tumor differentiation, mitotic count, as well as tumor necrosis [56]. The 8th edition is a notably improvement from the 7th edition, which failed to account for anatomic site, and spectrum of tumor sizes and therefore had limited applicability to RPS patient prognostication.

The Vanderbilt staging system is an alternative system that incorporates not only tumor size, but histologic grade and subtype, tumor extension, and multifocality. The system was validated with a cohort of 6857 patients from the National Cancer Database, and was found to have superior risk stratification with regards to 5-year overall survival [57]. However, this alternative staging system has not made their way into routine clinical practice.

In addition to stage, additional known prognostic factors include patient age at presentation, histologic subtype, complete resection with negative margins and without intra-operative tumor rupture, and provision of care at a specialized sarcoma center is associated with better outcomes. Therefore, various staging systems, nomograms and predictive tools have been devised [58,59,60,61].

The most notable nomograms are the Memorial Sloan Kettering nomogram and the Sarculator nomogram, both devised for the predictive limitations of the AJCC staging system [62,63]. Sarculator is, as of present, the best available tool for predicting overall survival and disease free survival in patients with primary resected retroperitoneal sarcomas [63,64]. Consequently, the nomogram has been made available for clinical use via the Sarculator app [65]. Additionally, a nomogram for outcome prediction in resected locally recurrent retroperitoneal sarcomas has been devised by the TARPSWG [66], while external validation studies are pending.

## 10. Multidisciplinary Care and International Collaborative

Treatment of RPS is challenging and complex. It is recommended that treatment be provided at high-volume designated tertiary sarcoma-care centers of excellence for the availability of a multidisciplinary care team, and strict adherence to clinical practice guidelines. It is well known that short- and long-term oncologic outcomes are improved at high-volume tertiary and multidisciplinary centers [54,67,68,69], despite the theoretical burden of travel time, distance and associated lack of social support away from home [70]. Although no international consensus exists regarding the number of retroperitoneal cases per year to yield a high-volume center, at least 11–20 cases per year is generally accepted [71]. In addition, such tertiary centers also provide access to participation in ongoing clinical trials. The Transatlantic Australasian Retroperitoneal Sarcoma Working Group (TARPSWG) is an international collaborative of at least 150 tertiary centers, designed to integrate data of this rare disease and produce consensus guidelines [72]. Creation of the global collaborative group has resulted in major advances in our understanding and in the management of this rare sarcoma subtype [72,73].

## 11. Management of Primary Retroperitoneal Sarcoma

### 11.1. Surgery for Locoregional Disease

Surgery is the mainstay and only true curative treatment for retroperitoneal sarcomas. Due to malnutrition at presentation, patients may require admission to hospital for general medical optimization which may include parenteral nutrition, pre-habilitation, and consultation with anesthesia.

The standardized technique of extended en-bloc resection of the retroperitoneal tumor with all adjacent organs was established by the Italian sarcoma group after showing significant oncologic improvement of local control [74,75]. The technique and associated details according to tumor side and structural involvement were presented at E-Surge Master Class in Sarcoma Surgery [76]. The technique entails a circumferential dissection from anterior to posterior in order to maintain vascular control, and good visualization with extended resection and peritoneal stripping in order for all surfaces of the tumor covered with healthy tissue [76]. The general approach involves a generous midline laparotomy. Tumors must be carefully studied for any extension through the diaphragm or pelvic canals, which would entail extension to a thoracoabdominal incision superiorly or an oblique incision inferiorly. This is typically followed by division of the gastrocolic ligament and of the transverse colon. Right-sided tumors may require a Cattell–Braasch maneuver (right medial visceral rotation) in order to assess involvement of the IVC, duodenum, head of the pancreas and ileum, and left-sided tumors may require a Mattox maneuver (left medial visceral rotation) to assess the distal pancreas, spleen, aorta with its branches, and rectum. In principle, en-bloc wide resection is required to obtain appropriate negative margins with >1 mm of microscopic tumor-free tissue (R0 resection). Adjacent organs which are commonly sacrificed with the primary tumor include the ipsilateral segmental colon, adrenal gland, kidney, psoas muscle and abdominal wall. With progression of disease outside the retroperitoneal plane, resection may include intraperitoneal bowel, spleen, and segmental liver. Although intraperitoneal invasion is associated with decreased overall survival, surgical resection should still be the mainstay of treatment in these patients [77]. Major vascular resection of the inferior vena cava, aorta and its major branches are intermittently required and indicated if reconstruction is possible [78]. Naturally, such benefit must be weighed against the risks of morbidity of resecting additional structures, such as the duodenum or pancreas necessitating pancreaticoduodenectomy, diaphragm or motor nerves, and kidney, all associated with possible significant morbidity [79,80,81,82]. Additional areas of concern preclude surgical resection include the superior mesenteric vessels, portal and hepatic vessels, pericardium and mediastinal structures, spinal cord and nephrectomy particularly if the contralateral kidney is non-functional [79,80,81,83,84]. Although surgical morbidity does impact patient quality of life and overall survival, there has been no association with local recurrence or distant metastasis according to a recent review [85]. In addition, careful consideration should be made for the histology subtype which may be associated with variable pattern of spread, local and distant recurrences, and consequently need for or against extensive resection [6,12,14]. Such surgical considerations for the most common subtypes of retroperitoneal sarcoma are outline in Table 3. RPS resection requires technical expertise of most abdominal and pelvic organs and vessels, and often requires collaboration with additional surgical teams.

### 11.2. Surgical Emergencies

Rarely does RPS present as a surgical emergency, but rather more indolent symptoms such as pain, bleeding, and gastrointestinal obstruction. Cases have been reported of spontaneous intraperitoneal tumor rupture with massive hemoperitoneum [86], lower gastrointestinal hemorrhage from duodenocaval fistula [87], intestinal perforation [88], gastric outlet obstruction [39], biliary obstruction and ureteric obstruction [89], among others. Overall, tumor rupture is a poor prognostic factor associated with high recurrence rate and poor survival likely related to underlying aggressive subtyping and subsequent peritoneal or systemic tumor spread.

### 11.3. Metastatic Disease

A small subset of patients present with synchronous distant metastasis (DM), with most common sites affected being the liver, the lung and the peritoneum in the form of sarcomatosis. DM are generally regarded as a contraindication for resection of primary RPS due to lack of a survival advantage, and are thus typically reserved in the setting of resectable metachronous disease [90].

### 11.4. Palliative Resection of Local Disease

The terms tumor ‘debulking’ or ‘partial reductive surgery’ have been applied to gross removal of locoregional tumor to alleviate symptomatology such as pain, obstruction and bleeding, and consequently improve quality of life. Tumor debulking is strictly considered a procedure with palliative intent in contrast to an incomplete R2 resection, where preoperative intent was rather curative. Studies have had mixed results with regards to survival advantage of reductive surgery in RPS, while considering that there are variations in tumor biology on survival even in the setting of R2 resection [91]. There are a limited number of studies which strictly look at the quality of life and oncologic benefit for palliative resections. Studies are on average small and fail to distinguish between patients with preoperative curative versus palliative intent surgery. Studies also fail to distinguish between patients with palliative endoscopic, interventional and operative procedures but rather encompass all into the category of “palliative procedures”. A study by Memorial Sloan Kettering Cancer Center in 2005 discovered an improvement in 70% of patients’ symptoms after palliative procedures with short term benefit of 130 days and tapering of symptom control to 54% at 100 days, and decrease thereafter. There was in general heightened procedural mortality of 12%, with significantly more complications within the operative group of 35% compared to the non-operative group of 4% [92]. A more recent 2020 US-sarcoma collaborative studied the outcomes of preoperative palliative intent resections in 70 primary and recurrent RPS patients. The authors discovered a heightened morbidity rate of 38% among all patients. Complications were significantly associated with incomplete R2 resection, tumor size >10 cm and low preoperative albumin levels. The authors found the presence of post-operative complication and high-grade tumor histology to be associated with a decreased overall survival, without any difference when accounting for the extent of resection (R0/1 versus R2, [93]). With limited oncologic advantage, and heightened morbidity and mortality, palliative surgery can be cautiously offered after careful selection and focused informed discussion.

## 12. Management of Recurrent Sarcoma

### 12.1. Locoregional Recurrence

In general, retroperitoneal sarcoma more frequently recurs locally as opposed to synchronous or only distant metastatic disease. Recurrences are typically asymptomatic and are discovered on routine surveillance, with the 5-, 8- and 10-year local recurrence rates for those with negative margins reported as 24%, 29.2% and 33.1%, respectively [9]. Thus, patients follow a close surveillance program, with variations according to tumor histology. Known predictors of recurrence include histologic subtype, multifocality at index surgery, organ invasion at index surgery, tumor growth rate, completeness of resection and management at a high-volume center [94,95,96,97]. In addition, recent research has demonstrated differentiation in tumor subtype of well-differentiate to or from dedifferentiated liposarcoma has implication on local recurrence and overall survival [98].

All patients are discussed at multidisciplinary care rounds at a specialized sarcoma center, with considerations made for multimodal therapy. Generally, if there are favorable tumor and patient factors [99], locoregional recurrence (LR) can be treated with re-resection following the same surgical principles as the index disease. This is especially true in cases of WDLPS where re-resection is potentially feasible with acceptable morbidity. It is rare that LMS will recur locally without any evidence of metastatic disease. All sites of metastatic disease must be ruled out prior to considering resection of a recurrent sarcoma of any histology. Moreover, in indolent WDLPS recurrences there may be a period of a “watch and wait” approach prior to considering resection. Other subtypes that may be considered for resection at the time of recurrence such as DDLPS, SFT and MPNST however this must be carefully balanced with the risk of substantial morbidity in which case alternative radiation or systemic treatments may be considered more appropriate.

Resectable recurrences have the benefit of extending survival, and may offer a “second chance” of cure in select patients [100]. However, the resectability rate of LR is reported as only 55%, largely due to the presence of multifocality and vascular invasion at the time of recurrence [95]. In addition, it is difficult to achieve R0 resections in LR with increased morbidity rates, with an incremental decrease in resection rates and increase with the number of recurrences. Symptomatic patients may thus be considered for palliative resection as outlined in the associated section of this review. An algorithm for management of recurrent RPS was published by Gyorki et al. [95] with a consensus approach for recurrent retroperitoneal sarcoma has been reached by the Transatlantic Australasian RPS Working Group [94].

### 12.2. Metastatic Disease

A small proportion of patients develop DM at the time of recurrence. Such recurrences are typically discovered on routine surveillance and need not always repeated biopsies should there be typical features on cross-sectional imaging. Regional therapy and metastastectomy in highly selected patients can confer survival advantage [91]. Typically, these are only considered after discussion at multidisciplinary case rounds in patients having undergone complete resection of the primary RPS, favorable biology, longer disease free interval between primary tumor surgery and DM, low volume and resectable disease, and sufficient ability to withstand such therapies [90,99].

## 13. Systemic Therapy

There is no current standard of adjuvant systemic therapy in retroperitoneal sarcoma. Systemic therapy in a neoadjuvant setting can theoretically cytoreduce the primary tumor and reduce the risk of distant metastasis. The argument against neoadjuvant therapy is its opportunity for delay of definitive surgical management, as well as local recurrence being more important than metastatic disease as the cause of mortality in this population. Retrospective reviews have shown conflicting results [101,102]. STRASS 2 (EORTC1809) is a randomized control trial which began accrual in 2020, tasked at evaluating the oncologic effects of neoadjuvant systemic chemotherapy in patients with RPS with estimated completion in 2028.

Some studies have demonstrated benefit of neoadjuvant systemic therapy for certain sarcoma subtypes, which have a predilection to chemosensitivity and who also metastasize hematologically. Although mostly extrapolated from data of extremity sarcomas, subtypes which may benefit from chemotherapy include Ewing’s sarcoma and embryonal rhabdomyosarcoma, both rare in the retroperitoneum but sensitive to chemotherapy, or for cytoreduction of high-grade dedifferentiated liposarcoma and leiomyosarcoma, both common in the retroperitoneum and may provide benefit to downstage tumors that are considered borderline resectable [102,103]. More recent work demonstrated not infrequent molecular differentiation at time of local recurrence [98] which may also impact decision-making for systemic therapies. It is known that metastatic patients can be treated with systemic therapy.

Studies analyzing the effect of adjuvant chemotherapy are largely based on extremity sarcomas, without any defined nor consistently reproductible results [102]. There is starting to be more data available regarding the oncologic benefits of combined systemic with radiation therapy, and combined systemic with regionalized hyperthermia [104], among others, are considered safe however not yet part of formal guidelines [54]. In addition, immunotherapy research sarcoma care is making considerable advancements, with bench and clinical data as well as ongoing randomized control trials outlined in a 2020 review [105].

## 14. Radiation Therapy

There is considerable variability in the timing, delivery and dosage of radiation therapy (RT) among patients with RPS. The rationale for radiation therapy as an adjunct to decrease the otherwise heightened risk of local recurrence, especially patients with positive surgical margins, high-grade tumors, and certain histologies.

With regards to timing, neoadjuvant radiation therapy is considered to improve the possibility of an otherwise difficult obtention of negative surgical margins. Intra-operative radiation and adjuvant therapy may additionally decrease the risk of local recurrence, while considering the toxic effects to surrounding kidneys, bowel and vessels and is thus rarely employed. Most research has explored the question regarding timing in the form of single-center retrospective reviews, until the development of the National Cancer Institute trial in 1993 [106]. This trial compared patients who underwent intra-operative compared to those who underwent adjuvant radiotherapy and found similar survival outcomes, with fewer local recurrences and complications among the intra-operative radiation group [106]; however, this was overall not practical in routine clinical practice [54]. Neoadjuvant radiation was explored in the ACOSOG-Z9031 (NCT00091351) trial however closed early due to poor accrual. This was followed by the multicenter randomized phase 3 STRASS-1 trial which also explored neoadjuvant radiation therapy compared to surgery alone, and reported no improvement with the addition of neoadjuvant radiotherapy to surgery on abdominal recurrence-free survival compared to surgery alone, for all RPS histologies [107]. Adjuvant therapy was previously felt to have a limited role in RPS; however a recent meta-analysis of 15 randomized control trials does allude to the possibility of decreased local recurrence, recurrence-free survival as well as overall survival [108].

Furthermore, data are beginning to suggest variations in response based on RPS subtypes [102]. Notably, in a subgroup analysis within the STRASS-1 trial, neoadjuvant radiation seemed to improve outcomes in patients with low-grade WDLPS but was not recommended in those with high-grade RPS [107]. A subsequent retrospective review of the National Cancer Database revealed a survival benefit only in those with high-grade liposarcoma measuring greater than 10 cm receiving radiation in the adjuvant setting [109]. Additional details regarding dosing, target volumes, techniques, and the overall role of RT in RPS management are outlined in a 2021 review [110], and summarized within the 2021 American Society of Radiation Oncology clinical practice guidelines for soft-tissue sarcoma [111]. Further studies are needed to address subgroups of RPS most likely to benefit from adjunctive radiation therapy.

## 15. Molecular Biology and Treatment Selection

The authors of a recent review outlined advancements in our understanding and treatments of sarcoma from a molecular biology point of view [112]. Notably, molecular profiling of STS has contributed to a greater understanding of this heterogenous disease and its subtypes. The authors comprehensively illustrate treatment regimens targeted at the cancer stem cells, cell signaling pathways, drug resistance pathways, extracellular stroma and matrix components, mesodermal tissues, perivascular components, and peritumoral and intra-tumoral microbiome, and immune cell tumor infiltrates. The progress in our understanding of the immune components of STS has led to the development of immune checkpoint inhibitors, activated NK cells, and CART T cells, also illustrated in this review. The authors have taken one step further to outline the cellular, molecular and metabolic biomarkers [113,114]. To further personalize care based on molecular profiling, many randomized control trials are underway. Notably, the MULTISARC (NCT03784014) is designed to randomize patients to undergo next generation sequencing of their tumor to then enter the phase II second-line targeted treatment to their genomic alternation. Furthermore, targeted therapies based on sarcoma subtype are outlined in a recent review [114].

## 16. Conclusions

Although there have been considerable advancements in the understanding and treatment of retroperitoneal sarcoma in 2021, standard treatment remains upfront primary surgical resection as the only potentially curative approach. Contemporary trials are investigating the role of neoadjuvant systemic therapy and immunotherapy, among others, will undoubtedly provide considerable insight into the treatment of this complex disease over the next decade.

## Figures and Tables

**Figure 1 cancers-14-01293-f001:**
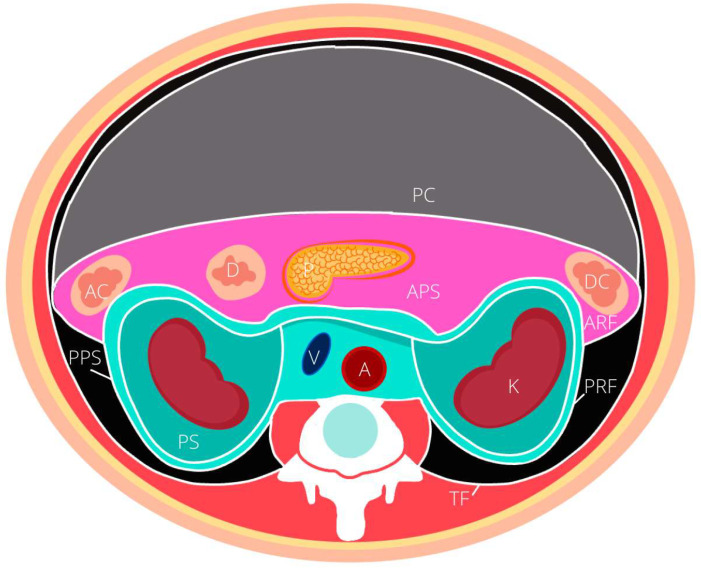
Schematic representation of the retroperitoneal perinephric compartments. ARF anterior renal fascia, PRF posterior renal fascia, TF transversalis fascia, APS anterior pararenal space, PPS posterior pararenal space, PS pararenal space, V vena cava, A abdominal aorta, D duodenum, AC ascending colon, DC descending colon, P pancreas, PC peritoneal cavity.

**Figure 2 cancers-14-01293-f002:**
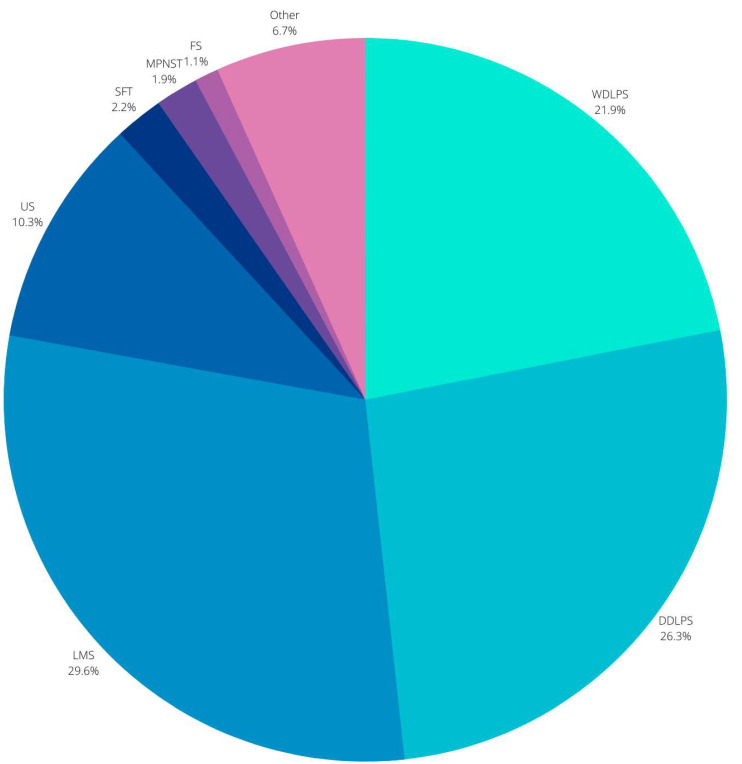
Diagram representation of the common subtypes of sarcoma within the retroperitoneum: DDLPS dedifferentiated liposarcoma; WDLPS well-differentiated liposarcoma, LMS leiomyosarcoma; US undifferentiated sarcoma; SFT solitary fibrous tumor, MPNST malignant peripheral nerve sheath, FS fibrosarcoma [9].

**Table 1 cancers-14-01293-t001:** Retroperitoneal compartments with associated spaces, borders and contents.

Compartment	Space	Borders	Contents
Greater Vessel	Greater Vessel Space	Superior: posterior mediastinum Posterior: vertebral bodies, psoas muscle Lateral: perirenal spaces and ureters	Abdominal aorta Inferior vena cava Lymphatics
Posterior	Posterior Space	Anterior: transversalis fascia	Psoas muscle
Lateral	Anterior Pararenal Space (APS)	Anterior: Parietal peritoneum and intraperitoneal space Posterior: anterior renal fascia and perirenal space	Pancreas head and neck Duodenum (parts 2–4) Ascending Colon Descending Colon
	Perirenal Space (PS)	Superior: diaphragm Anterior: Gerotas fascia and anterior pararenal space Posterior: Zuckerkandl fascia and posterior pararenal space	Adrenal gland Kidney Renal hilum with ureter, artery and vein
	Posterior Pararenal Space (PPS)	Lateral: lateroconal fascia Anterior: posterior renal fascia and perirenal space Posterior: Transversalis fascia	No major organs Fat pad ventral to quadratus lumborum

**Table 2 cancers-14-01293-t002:** Genetic syndromes associated with the development of soft-tissue sarcomas.

Familial Syndromes	Gene Mutation (Chromosome)	Associated Sarcomas
Neurofibromatosis type I (Von Recklinghausen’s disease)	NF1 (17q11.2)	GIST, RMS, MPNST
TP53-related cancer syndrome (Li–Fraumeni)	TP53 (17q13.1, 1q23), CHEK2 (22q12)	RMS, LMS, LPS, FHT [22]
Hereditary retinoblastoma	RB1 (13q14)	Bone and STS [23]
Familial rhabdoid predisposition syndrome I	SMARB1/INI1 (22q11.33)	MRT
Familial rhabdoid predisposition syndrome II	SMARCA4 (19q13.2)	MRT
Hereditary leiomyomatosis and renal cell cancer	FH (1q42)	Uterine LMS
Multiple osteochrondromas	EXT1 (8q24), EXT2 (11p11)	Chrondrosarcoma
Rubinstein–Taybi	CREBBP (16p13.1)	LMS
Tuberous sclerosis	TSC1 (9q34), TSC2 (16p13.3), TSC3 (12q22–24.1)	PEComa, chordoma
Mafucci syndrome	IDH1 (2q34), IDH2 (15q26.1)	Angiosarcoma, chondrosarcoma
Nijmegen breakage syndrome	NBN (8q21.3)	RMS

GIST gastrointestinal stromal tumors; MPNST malignant peripheral nerve sheath tumor; RMS rhabdomyosarcoma; LMS leiomyosarcoma; LPS liposarcoma; FHT fibrohistocytic tumor; MRT malignant rhabdoid tumor; STS soft-tissue tumors; PEComa perivascular epithelioid cell differentiation tumors.

**Table 3 cancers-14-01293-t003:** Retroperitoneal sarcoma subtypes with associated patten of spread, mechanisms of local and distant failure with associated 5-year local recurrence and 5-year metastatic disease, and surgical implications.

Sarcoma Subtype	Pattern of Spread	Mechanism of Failure (5-Year %)	Surgical Implications
WDLPS	Adipose infiltration Multilobulated Indistinct borders	LR (19–39%) >> MD (0%)	Extended en-bloc resection requiring ipsilateral retroperitoneal fat resection
DDLPS	Adipose and Visceral infiltration Multilobulated Indistinct borders	G2: LR (44%) > MD (10%) G3: LR (33%) << MD (44%)	Extended en-bloc resection requiring ipsilateral retroperitoneal fat resection
LMS	Distinct borders	LR (6–16%) << MD (55–56%)	En-bloc resection with vascular structures May preserve adjacent critical structures
MPNST	Distinct borders	LR (20–35%) > MD (12–13%)	En-bloc resection with associated neurovascular structures
SFT	Distinct borders	LR (4–8%) > MD (17%)	En-bloc resection May preserve adjacent critical structures

WDLPS well-differentiated liposarcoma; DDLPS dedifferentiated liposarcoma, G2 grade 2, G3 grade 3, LMS leiomyosarcoma, MPNST malignant peripheral nerve sheath tumor, SFT solitary fibrous tumors [6,12,14].

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
