# Peer review of "Retroperitoneal Sarcoma Care in 2021"

_cancers, 2022, doi:10.3390/cancers14051293_

Round 1
Reviewer 1 Report
Manuscript ID: 1588885
Title: Retroperiotoneal Sarcoma Care in 2021
In this review, Schmitz and Nessim provide a thorough review of the management of RP sarcomas. I have a few comments:
- On page 24, line 38, I would caution the authors in highlighting that advancement of RP sarcoma care is “largely due” to 1 particular group. This sentence needs to be softened or more inclusive as there are other significant contributions, not related to TARPSWG alone.
- Line 64, “do not often invade” is too strong. RP tumors can invade adjacent organs. WDLPS LPS often push but DDLPS are more invasive, as are SFT. This should be revised.
- Line 117: I think there is a word or colon missing after frequency
- Line 121: there is a ? after fibrosarcoma
- Line 222: Althought the medican latency is proabably 5 years at least, radiation-associated sarcoma can occur before 5 years, would soften the sentence. https://pubmed.ncbi.nlm.nih.gov/25743327/
- Line 319: sarculator is not in the staging system but in the recommended section. Please update
Author Response
Please see attached document with the point-by-point responses to the reviewer's comments.

Reviewer 2 Report
Genetics, and Familial Syndromes are not important in retroperitoneal sarcomas, because they have little importance in this setting of patients. Instead, I would consider describing the news regarding molecular biology that should help in choosing the treatment.
The role of surgery after recurrence needs to be better explained, especially in relation to the different histologies.
The paper reports that "Solitary fibrous tumors were historically the most common entity within the retroperitoneum"....., it is a mistake. I think the authors mean malignant fibrous histiocytomas and not SFT.
Finally, I would recommend to the authors to interpret the data and assess it in a critical fashion.
Author Response

(The authors gave the same response as above.)

Reviewer 3 Report
This paper is clear and easy to follow and gives a good overview of the current status of retroperitoneal sarcoma.
The pattern of spread in Table 3 describes WDLS and DDLPS as “multifocality”.
However, the meaning of “multifocality” in this paper is not clear. “Multifocal” generally means multiple tumors in a discontinuous manner, and the authors have mentioned that WDLS shows an infiltrative growth pattern. Even if tumors appear to be multifocal, there is a possibility that the tumors are in continuity in some areas, and therefore, the term “multifocal” should not be used.
Author Response

(The authors gave the same response as above.)

Round 2
Reviewer 2 Report
Accept in present form